# Phenolic Constituents from *Wendlandia tinctoria* var. *grandis* (Roxb.) DC. Stem Deciphering Pharmacological Potentials against Oxidation, Hyperglycemia, and Diarrhea: Phyto-Pharmacological and Computational Approaches

**DOI:** 10.3390/molecules27185957

**Published:** 2022-09-13

**Authors:** Mamtaz Farzana, Md. Jamal Hossain, Ahmed M. El-Shehawi, Md. Al Amin Sikder, Mohammad Sharifur Rahman, Muhammad Abdullah Al-Mansur, Sarah Albogami, Mona M. Elseehy, Arpita Roy, M. Aftab Uddin, Mohammad A. Rashid

**Affiliations:** 1Department of Pharmaceutical Chemistry, Faculty of Pharmacy, University of Dhaka, Dhaka 1000, Bangladesh; 2Department of Pharmacy, State University of Bangladesh, 77 Satmasjid Road, Dhanmondi, Dhaka 1205, Bangladesh; 3Department of Biotechnology, College of Science, Taif University, P.O. Box 11099, Taif 21944, Saudi Arabia; 4Institute of National Analytical Research and Service (INARS), Bangladesh Council of Scientific and Industrial Research (BCSIR), Dr. Qudrat-I-Khuda Road, Dhanmondi, Dhaka 1205, Bangladesh; 5Department of Genetics, Faculty of Agriculture, University of Alexandria, Alexandria 21545, Egypt; 6Department of Biotechnology, School of Engineering & Technology, Sharda University, Greater Noida 201310, India; 7Department of Genetic Engineering and Biotechnology, Faculty of Biological Sciences, University of Dhaka, Dhaka 1000, Bangladesh

**Keywords:** *Wendlandia tinctoria*, phytochemical isolation, polyphenols, antioxidant, hypoglycemic, antidiarrheal, molecular docking

## Abstract

*Wendlandia tinctoria* var. *grandis* (Roxb.) DC. (Family: Rubiaceae) is a semi-evergreen shrub distributed over tropical and subtropical Asia. The present research intended to explore the pharmacological potential of the stem extract of *W. tinctoria*, focusing on the antioxidant, hypoglycemic, and antidiarrheal properties, and to isolate various secondary metabolites as mediators of such activities. A total of eight phenolic compounds were isolated from the dichloromethane soluble fraction of the stem extract of this plant, which were characterized by electrospray ionization (ESI) mass spectrometric and ^1^H NMR spectroscopic data as liquiritigenin (**1**), naringenin (**2**), apigenin (**3**), kaempferol (**4**), glabridin (**5**), ferulic acid (**6**), 4-hydroxybenzoic acid (**7**), and 4-hydroxybenzaldehyde (**8**). The dichloromethane soluble fraction exhibited the highest phenolic content (289.87 ± 0.47 mg of GAE/g of dried extract) and the highest scavenging activity (IC_50_ = 18.83 ± 0.07 µg/mL) against the DPPH free radical. All of the isolated compounds, except 4-hydroxybenzaldehyde, exerted a higher antioxidant effect (IC_50_ = 6.20 ± 0.10 to 16.11 ± 0.02 μg/mL) than the standard butylated hydroxytoluene (BHT) (IC_50_ = 17.09 ± 0.01 μg/mL). Significant hypoglycemic and antidiarrheal activities of the methanolic crude extract at both doses (200 mg/kg bw and 400 mg/kg bw) were observed in a time-dependent manner. Furthermore, the computational modeling study supported the current in vitro and in vivo findings, and the isolated constituents had a higher or comparable binding affinity for glutathione reductase and urase oxidase enzymes, glucose transporter 3 (GLUT 3), and kappa-opioid receptor, inferring potential antioxidant, hypoglycemic, and antidiarrheal properties, respectively. This is the first report of all of these phenolic compounds being isolated from this plant species and even the first demonstration of the plant stem extract’s antioxidant, hypoglycemic, and antidiarrheal potentials. According to the current findings, the *W. tinctoria* stem could be a potential natural remedy for treating oxidative stress, hyperglycemia, and diarrhea. Nevertheless, further extensive investigation is crucial for thorough phytochemical screening and determining the precise mechanisms of action of the plant-derived bioactive metabolites against broad-spectrum molecular targets.

## 1. Introduction

Medicinal plants have been used for millennia to treat human ailments and maintain health worldwide. Natural products and alternative therapies have received increasing attention throughout the years because of the adverse effects of conventional medicines and the growing demand for more natural products free of toxins [1,2,3]. Through experimental and clinical investigations, the use of specific natural products and herbal extracts for managing human diseases and disorders has finally begun to receive scientific justification [4,5]. According to the World Health Organization (WHO), in most developing countries, particularly those in Asia, Africa, Latin America, and the Middle East, 70–95% of the population use traditional medicine including herbal medicines to address their health care needs and concerns [6,7,8]. Due to the scarcity of contemporary medical facilities, the efficacy of traditional medicines, and cultural interests and preferences, natural products provide a low-cost alternative for primary health care in developing nations [9,10].

*Wendlandia tinctoria* var. *grandis* (Roxb.) DC. is a semi-evergreen hermaphroditic flowering plant. The genus *Wendlandia* belongs to the family Rubiaecae with over 90 diverse species found in tropical and subtropical regions of Asia and one species in Africa [11]. During summer, *W. tinctoria* serves as a key nectar source for butterflies and as a pollen source for honey bees [12]. Among tribal communities, *W. tinctoria* is commonly used as an antidote to treat snakebite. Bark of this plant is used to relieve cramps in cholera patients [13]. According to a previous report, the stem of *W. tinctoria* contains geniposidic acid, myricyl stearate, stearic acid, d-mannitol, β-sitosterol, and stigmasterol [14]. The root of *W. tinctoria* was found to contain various iridoid glucosides, namely wendoside, 8-epi-mussaenoside, 5-dehydro-8-epi-mussaenoside, 5-dehydro-8-epi-adoxosidic acid, and 10-O-dihydro-feruloyldeacetyldaphylloside [14,15].

The progression of degenerative and chronic diseases, including cancerous growths, diabetes, immunological issues, joint inflammation, and cardiovascular and neurological disorders has been significantly linked to oxidative stress [16]. The production of ROS by lipoxygenases, cyclooxygenases, nitric oxide synthases, and peroxidases is assumed to be triggered by hyperglycemia in diabetics [17]. The substantial antioxidant activity of phytocompounds, especially polyphenols, has been demonstrated to prevent oxidative stress by reducing ROS generation and retaining redox homeostasis [18,19,20]. The latest conventional anti-diabetic medications may alleviate diabetes symptoms, but they also carry a risk of severe adverse reactions, including hypoglycemic shock, edema, irregular heartbeat, and liver function as well as stomach and respiratory problems [21]. As a result, scientists are working to identify and characterize polyphenols with potent antioxidant properties in medicinal plants, which can be used as alternatives to traditional diabetes therapies because of their lower toxicity and significant antioxidative properties [21,22].

Since worldwide phytochemical research has resulted in the isolation of a myriad of secondary metabolites with miscellaneous pharmacological properties, plant-derived bioactive compounds are considered as a crucial source of lead compounds from which novel therapeutic agents can be developed in many major disease areas [23,24]. No previous study in the literature has reported on the antioxidant, hypoglycemic and antidiarrheal activity of *W. tinctoria*. Although many studies have been carried out on several species of the genus *Wendlandia*, phytochemical studies on the genus *W. tinctoria* are extremely limited. Therefore, this research aimed to explore the in vitro antioxidant and in vivo hypoglycemic and antidiarrheal potential of *W. tinctoria* stem, including phytochemical screening for chemically distinct and structurally diverse secondary metabolites from this plant species. Furthermore, in order to predict and support the biological activities of the isolated compounds, molecular docking was also applied. Thus far, this is the first report on the isolation of several potential phenolic compounds from this plant species as well as the first-time demonstration of the antioxidant, hypoglycemic, and antidiarrheal potential of *W. tinctoria* stem, which may be considered as a prospective natural source for managing oxidative stress, hyperglycemia, and diarrhea.

## 2. Materials and Methods

### 2.1. Plant Material

In January 2020, the aerial portion of *Wendlandia tinctoria* var. *grandis* (Roxb.) DC. was collected from Sylhet, Bangladesh’s north-eastern mountainous region. A specialist from the Bangladesh National Herbarium (BNH), Mirpur, Dhaka, validated the plant’s taxonomical identification. The herbarium received a voucher specimen, and DACB 56484 was assigned as its accession number.

### 2.2. Reagents and Chemicals

Analytical grade reagents and chemicals were used in this research work. All of the solvents, gallic acid, Folin–Ciocalteau reagent, tert-butyl-1-hydroxytoluene (BHT), and Tween 80 were bought from Merck (Darmstadt, Germany). 2,2-Diphenyl-1-picrylhydrazyl (DPPH) was purchased from Sigma-Aldrich, St. Louis, MO, USA. Normal saline solution, glibenclamide, and loperamide were collected from Square Pharmaceuticals Ltd., Dhaka, Bangladesh.

### 2.3. Extraction

The air-dried and finely ground stem powder (500 g) was subjected to maceration with 4 L methanol in an airtight amber-colored bottle for 15 days with periodic shaking. After that, filtration of the solvent mixture was conducted through a fresh cotton plug and then by filter paper. Finally, the filtrate was concentrated by a rotary evaporator (Buchi Rotavapor; BUCHI Labortechnik AG, Flawil, Switzerland) at a temperature around 40–50 °C and under reduced pressure to obtain a viscous crude extract. The percent yield of the solid crude methanolic extract of the stem was found to be 67.2%.

### 2.4. Fractionation

The methodology designed by S. Morris Kupchan (1970) and modified by VanWagenen et al. [25] was used for solvent-solvent partitioning of the crude extract. For this purpose, 10 gm of the crude extract was dissolved in a 10% aqueous methanol and was extracted sequentially with n-hexane, dichloromethane (DCM), ethyl acetate, and distilled water to obtain four respective fractions. Finally, all of the fractions were evaporated to dryness. The percent yield of the n-hexane, dichloromethane, ethyl acetate, and aqueous soluble fractions was 25%, 34%, 18%, and 20%, respectively.

### 2.5. Chromatographic Procedures

The dichloromethane soluble fraction was selected for the isolation of secondary metabolites and hence, was further fractionated by size exclusion chromatography using a gradient solvent system as the mobile phase and lipophilic sephadex LH 20 as the stationary phase. The ratio of different solvents used as the mobile phase was as follows—n-hexane:DCM:methanol (2:5:1), 10% methanol in DCM, 20% methanol in DCM, 50% methanol in DCM, 80% methanol in DCM, and 100% methanol (Appendix A). The thin layer chromatography (TLC) technique was employed for extensive screening of these fractions using different solvent systems over aluminum plates (20 × 20 cm) coated with silica gel (Kieselgel 60 F_254_). The developed chromatogram was visually checked under UV light at 254 nm and 366 nm for the detection of fluorescence quenching. Then, the plate was observed carefully after the application of 1% vanillin-sulfuric acid spray followed by heating at 105 °C for 2–3 min for marking the presence of colored compounds. Fractions showing a similar TLC pattern were mixed together, and consequently, 31 sub-fractions were obtained. For the isolation and purification of the compounds from these sub-fractions, the repeated preparative thin layer chromatography (PTLC) technique was performed.

A total of eight compounds (**1**–**8**; Figure 1) were isolated in the study. Compound **1** was isolated from sub-fraction F-25 (using a solvent mixture of 55% ethyl acetate in toluene), compound **2** from sub-fraction F-26 (using 50% ethyl acetate in toluene), compound **3** from sub-fraction F-27 (using 55% ethyl acetate in toluene), compound **4** from sub-fraction F-30 (using 60% ethyl acetate in toluene), compound **5** from sub-fraction F-24 (using 55% ethyl acetate in toluene), compound **6** from sub-fraction F-23 (using 50% ethyl acetate in toluene), compound **7** from sub-fraction F-29 (using 55% ethyl acetate in toluene), and compound **8** from sub-fraction F-28 (using 50% ethyl acetate in toluene) of the dichloromethane soluble fraction of the stem extract of *W. tinctoria* (Appendix A). Appendix A depicts the schematic diagram of the whole phytochemical isolation.

### 2.6. Experimental Procedures

The ^1^H NMR spectra were recorded on a Bruker AMX-400 NMR spectrometer at 400 MHz. All of the spectra were acquired in a deuterated solvent (CD_3_OD) and the chemical shift (*δ*) values were referenced to tetramethylsilen (TMS) and the residual solvent signals. The mass spectra were recorded on an AB SCIEX 3200MD QTRAP mass spectrometer by the electrospray ionization (ESI) technique.

### 2.7. Experimental Animals

The 4–5 week-old, 25–30 g healthy adult Swiss albino mice of both sexes were obtained from the animal division of the International Center for Diarrheal Diseases and Research in Bangladesh (ICDDR,B). Before beginning the investigation on animals, the mice were housed in polypropylene cages under typical environmental conditions of 25 ± 2 °C ambient temperature, 55 ± 10% relative humidity, and a 12 h light/12 h dark cycle while receiving ICDDR,B prepared mice food and water [26]. While conducting the research, the criteria and recommendations of the Federation of European Laboratory Animal Science Associations (FELASA) were meticulously complied with the adopted protocols of the animal study. Furthermore, the Animal Ethics Committee of the State University of Bangladesh has critically reviewed and approved the study’s complete ethical guidelines and protocols (Approval Number: 2021-05-25/SUB/A-ERC/003). A total of 32 mice were used in the investigation, 16 of which were used in the antidiarrheal experiment and the other 16 were used in the hypoglycemic test. There were four groups in each test as follows:Group I = Negative Control Group (CTL);Group II = Positive Control Group (STD);Test Group I = Methanolic Crude Extract of *W. tinctoria* at 200 mg/kg bw (MCE 1);Test Group II = Methanolic Crude Extract of *W. tinctoria* at 400 mg/kg bw (MCE 2).

### 2.8. Antioxidant Effects

#### 2.8.1. Determination of Total Phenolic Content (TPC)

By using the Folin–Ciocalteu assay, the stem extractives’ total phenolic content was determined [27]. A total of 2.5 mL of Folin–Ciocalteu reagent (diluted ten-fold with distilled water) and 2.5 mL of NaCO_3_ (7.5% *w*/*v*) solution were added to 1 mL of each stem extractive (1 mg/mL). The mixture was incubated for 30 min at room temperature and in a dark area. A UV–Vis spectrophotometer was used to detect the absorbance at 760 nm. The gallic acid standard calibration curve was prepared by plotting the absorbance against different concentrations (250, 125, 62.5, 31.25, and 15.625 µg/mL) of gallic acid solutions to quantify the total phenolic content of the test samples expressed in terms of mg of GAE (gallic acid equivalent)/g of the dried extractives. The TPC test was carried out three times. Equations of the calibration curve of gallic acid were as follows:1st experiment: y = 0.0123X − 0.19; R^2^ = 0.9979
2nd experiment: y = 0.0137X − 0.2235; R^2^ = 0.9977
3rd experiment: y = 0.014X − 0.241; R^2^ = 0.9978

#### 2.8.2. DPPH Scavenging Assay

By using the DPPH assay, the antioxidant capacity of the stem extractives as well as the isolated compounds was quantitatively assessed [28]. First, 2 mL of each methanolic solution of the test materials of different concentrations (250, 125, 62.5, 31.25, and 15.625 µg/mL) obtained by serial dilution were properly mixed with 3 mL of the newly prepared methanolic DPPH solution (0.004% *w*/*v*), followed by incubation for 30 min in the dark at room temperature. The absorbance was then measured using a UV–Vis spectrophotometer at 517 nm against methanol as the blank. The percentage of inhibition (I%) of the DPPH free radical was determined with the following formula [29]:% Inhibition = (A_control_ − A_sample_)/A_control_ × 100%
where A_control_ is the absorbance of the solution that contains all reagents, except the test material. A_sample_ is the absorbance of the solution containing the test material or standard (BHT). A graph of % of inhibition vs. the sample concentrations was constructed for each test material, and finally, linear regression analysis was used for the calculation of IC_50_ (the half maximal inhibitory concentration) values.

### 2.9. Hypoglycemic Activity

Hypoglycemic activity of *W. tinctoria* was evaluated by the oral glucose tolerance test in Swiss albino mice [30]. All of the mice were kept in the fasting state for 18 h before conducting the experiment. First, the blood glucose level of each mouse was measured at zero hour by a one touch glucometer (Bioland G-423S). Then, a particular treatment was administered to each group orally. A total of 1% Tween 80 in normal saline solution at a dose of 10 mL/kg of body weight was administered to the negative control group and the standard drug glibenclamide at a dose of 10 mg/kg of body weight was administered to the positive control group. The two test groups MCE 1 and MCE 2 received the methanolic crude extract at doses of 200 and 400 mg/kg of body weight, respectively. One hour later, each group received 10% glucose solution (2 mg/kg of body weight) orally. At the first, second, and third hour after glucose loading, blood was withdrawn from the tail vein of each mouse to measure the blood glucose level for the evaluation of the hypoglycemic effect of the test samples compared to the negative and positive control groups. The hypoglycemic activity was expressed in terms of % reduction in the blood glucose level. The following formula was used to calculate the % reduction in the blood glucose level:% Reduction in blood glucose level = (A − B)/A × 100%
where A is the mean blood glucose level of the negative control group and B refers to the mean blood glucose level of the positive control group or the test groups.

### 2.10. Antidiarrheal Activity

The antidiarrheal activity of *W. tinctoria* was assessed through the castor oil-induced diarrhea method in mice described by Shoba and Thomas [31]. All experimental animals were fasted for 18 h prior to the experiment. The treatment and dose of the negative control group and the test groups were followed as mentioned in Section 2.9. Standard drug loperamide was administered orally to the positive control group at a dose of 50 mg/kg of body weight. After 1 h, each mouse was fed 1 mL pure castor oil for inducing diarrhea. For each mouse, a separate cage was allocated with blotting paper on the floor. The number of diarrheal feces for each mouse was noted at the end of each hour for up to four hours after castor oil administration. At the start of each hour, the blotting paper was changed. To determine the antidiarrheal potential of *W. tinctoria*, the observations of the test groups were compared to that of the negative and positive control groups. The antidiarrheal activity was evaluated by calculating the % reduction of diarrhea using the formula below:% Reduction of diarrhea = (A − B)/A × 100%
where A is the mean number of diarrheal feces of the negative control group and B is the mean number of diarrheal feces of the positive control group or test groups.

### 2.11. Molecular Docking Study

Computational modeling analysis of the isolated compounds from *W. tinctoria* and the selected standard drugs against their target proteins was performed in order to anticipate the receptor binding profile of these compounds. The molecular docking intervention of these phytocompounds was carried out using PyRx, PyMoL 2.3, and BIOVA Discovery Studio version 4.5 following the methods described in several studies [32,33,34,35,36,37].

#### 2.11.1. Protein Selection

The identified and isolated compounds’ potential antioxidant, antidiarrheal, and hypoglycemic characteristics were interpreted using molecular docking. The proteins kappa opioid receptor (PDB ID: 6VI4), urase oxidase (PDB ID: 1R4U), and glutathione reductase (PDB ID: 3GRS), and glucose transporter 3 [GLUT 3] (PDB ID: 4ZWB) were selected to conduct the molecular interaction and determine the antidiarrheal, radical scavenging ability, and hypoglycemic potentialities, respectively, based on the previously published evidence [35,38]. The RCSB protein data bank (https://www.rcsb.org; accessed on 1 June 2022) was used to obtain the three-dimensional (3D) crystal structures of the chosen proteins, which were then downloaded in PDB format. In order to eliminate any water molecules and undesirable protein residues, all biomolecules were opened with the PyMoL 2.3 program. Using a Swiss PDB viewer and an energy minimization tool, non-polar hydrogen atoms were added to the cleaned proteins to assemble them, and they were then changed to their lowest energy state [39]. The cleaned and optimized macromolecules were then saved in PDB format for upcoming analysis.

#### 2.11.2. Ligand Preparation

Figure 1 depicts the structural representation of all of the isolated phytocompounds (**1**–**8**). These eight compounds were found in the PubChem database (https://pubchem.ncbi.nlm.nih.gov/; accessed on 1 June 2022) as liquiritigenin (PubChem CID: 114829), naringenin (PubChem CID: 932), apigenin (PubChem CID: 5280443), kaempferol (PubChem CID: 5280863), glabridin (PubChem CID: 124052), ferulic acid (PubChem CID: 445858), 4-hydroxybenzoic acid (PubChem CID: 135), and 4-hydroxybenzaldehyde (PubChem CID: 126), respectively. The ligands and standard drugs’ 3D conformers, antioxidant BHT (PubChem CID: 31404), loperamide (PubChem CID: 3955), and glibenclamide (PubChem CID: 3488), were retrieved and secured in SDF format. A ligand library in PDB format was created using the PubChem CIDs of the ligands after they had been serially loaded into Discovery Studio version 4.5. The Pm6 semi-empirical method was used to improve the accuracy of the molecular interaction for all phytoconstituents and standard ligands [40].

#### 2.11.3. Ligand and Protein Interaction

The potential binding profiles of the isolated plant metabolites with their binding affinities toward the target biomolecules were assumed using molecular docking [41]. A semiflexible modeling strategy was used during the computational-aided interaction process, which was conducted using the commonly employed advanced software PyRx AutoDock Vina for the molecular docking. The desired protein was loaded and designated as a macromolecule. The amino acids with three-letter IDs from the literature were chosen to determine the site-specific ligand–protein interaction. A total of eleven amino acids, VAL 102, LYS 127, ASN 129, VAL 130, GLN 131, LYS 143, SER 145, SER 147, GLY 148, ASP 183, and THR 185 had been selected in the glutathione reductase enzyme (PDB ID: 3GRS) and fourteen amino acids viz. VAL 73, THR 74, PRO 76, PRO 75, ARG 128, TRP 208, TRP 106, THR 107, ARG 105, HIS 104, CYS 103, MET 32, GLU 31, and TYR 30 have been chosen in urase oxidase (PDB ID: 1R4U) for conducting lively site-specific docking to predict the antioxidant outcomes of these isolated phytoconstituents [38]. LEU 103, LEU 107, SER 136, ILE 137, TRY 140, ILE 180, TRP 183, LEU 184, SER 187, ILE 191, LEU 192, ILE 194, and VAL 195 were identified while selecting active sites of the kappa opioid receptor (PDB ID: 6VI4) to dock with the isolated phytocompounds for predicting antidiarrheal potentiality [35]. The active sites of the GLUT3 (PDB ID: 4ZWB) proteins were also chosen from the literature [35]. All of the ligands’ 3D conformers (SDF format) were uploaded into the PyRx software and operated for energy minimization. To integrate the most suitable optimal hit, all ligands were converted into pdbqt format in the PyRx AutoDock Vina software using the Open Babel tool. The grid box was then created, and the active binding sites of the proteins were kept within the center of the box, where the grid box mapping was as follows for the center (X, Y, Z): (50.4368, 48.1950, 12.3616) and dimensions (angstrom): (37.0910, 28.8059, 45.1400) during interacting with the 3GRS protein. For the 1R4U protein, the grid box mapping was set at the following coordinates: center (X, Y, Z): (23.1360, 57.9585, 44.3323, and dimensions (X, Y, Z): (16.7970, 19.7225, 22.0017), in contrast to the center grid box mapping for the 6VI4 protein, which was (X, Y, Z): (54.5401, −50.8237, −16.2425) for the center mapping and dimensions (angstrom) (X, Y, Z): (16.4228, 28.0026, 18.5286). Corresponding to this, the grid box’s center space was maintained as the location of each receptor’s active binding sites, and grid box mapping values were noted. The rest of the docking process’s parameters were left at their default values. Docking was carried out using AutoDock Vina under all indicated circumstances (version 1.1.2; Scripps Research Institute, San Diego, CA, USA). The docked macromolecules and ligands were exported as out files (pdbqt format), and the docking analysis results were projected. For further visualization, the out files of the ligands and the pdbqt file of the macromolecule were merged and saved into PDB format using PyMol software. The visualization and creation of 2D figures were conducted using Discovery Studio Visualizer (version 4.5; BIOVIA, San Diego, CA, USA).

### 2.12. Statistical Analysis

In vitro antioxidant studies were carried out in triplicate (*n* = 3) and the findings were presented as mean ± SEM (standard error of mean). The findings of the in vivo studies are expressed as mean ± SEM (*n* = 4) and percentages. GraphPad Prism (version 9.3.1; GraphPad Software Inc., San Diego, CA, USA) was used to analyze the data by one-way ANOVA and Dunnett’s multiple comparison test. In comparison to the negative control group, the observed values were regarded as statistically significant at * *p* ˂ 0.01, ** *p* ˂ 0.001, and *** *p* ˂ 0.0001. Molecular docking was also conducted in triplicate and the error of the test was less than 1%.

## 3. Results

### 3.1. Phytochemical Studies

In the current research, a total of eight compounds were isolated from the dichloromethane soluble fraction of the stem extract of *W. tinctoria*, which included liquiritigenin (**1**), naringenin (**2**), apigenin (**3**), kaempferol (**4**), glabridin (**5**), ferulic acid (**6**), 4-hydroxybenzoic acid (**7**), and 4-hydroxybenzaldehyde (**8**) (Figure 1). Structures of the isolated compounds were elucidated by extensive analysis of ESI mass and ^1^H NMR spectroscopic data as well as by comparative study against previously reported data (Table 1).

Compound **1** was obtained as a lemon yellow crystalline solid. The ESI-MS spectral data exhibited a pseudo-molecular ion peak at m/z 257, corresponding to [M + H]^+^, in accordance with a molecular formula C_15_H_12_O_4_. The ^1^H NMR spectrum demonstrated typical ABX splitting pattern for a flavanone with resonances at δ 5.36 (1H, dd, *J* = 2.8 and 13.2 Hz, H-2), 3.04 (1H, dd, *J* = 13.2 and 16.6 Hz, H_a_-3), and 2.67 (1H, dd, *J* = 2.8 and 16.6 Hz, H_b_-3). The spectrum also demonstrated two doublets at δ 7.72 (1H, *J* = 8.0 Hz, H-5) and 6.34 (1H, *J* = 2.0 Hz, H-8) and a doublet of doublets at δ 6.48 (1H, *J* = 2.0 and 8.0 Hz, H-6), which altogether indicated ring A of the flavanone skeleton. In addition, two doublets at δ 7.33 (2H, *J* = 8.1 Hz, H-2′ and H-6′) and 6.80 (2H, *J* = 8.1 Hz, H-3′ and H-5′) indicated the presence of para-disubstituted aromatic ring at C-2. In accordance with these spectral data being compared with the published data [42], compound **1** was identified as 4′,7-dihydroxyflavanone, also known as liquiritigenin (Table 1).

Compound **2** was isolated as a colorless crystalline solid. The ESI-MS spectral data demonstrated a pseudo-molecular ion peak [M + H]^+^ at m/z 273, compatible with the molecular formula C_15_H_12_O_5_. The ^1^H NMR spectrum exhibited three doublets of doublets at δ 5.35 (1H, *J* = 13.2, 2.4 Hz, H-2), 2.69 (1H, *J* = 17.2 and 2.8 Hz, H_a_-3), and 3.10 (1H, *J* = 17.2 and 13.2 Hz, H_b_-3), indicating a typical ABX type splitting pattern, characteristic of a flavanone. The spectrum also displayed two broad singlets at δ 5.87 and 5.88, which accounted for the H-6 and H-8 protons, respectively, indicative of ring A of the flavanone skeleton. Furthermore, two doublets at δ 6.80 (2H, *J* = 8.00 Hz, H-2′ and H-6′) and 7.40 (2H, *J* = 8.0 Hz, H-3′ and H-5′) demonstrated the presence of a 4′-oxygenated phenyl moiety at C-2. A comparison of these spectral data with the reported values [43] confirmed the structure of compound **2** as 4′,5,7-trihydroxyflavanone, also named as naringenin (Table 1).

Compound **3** was isolated as a yellow pale crystalline powder. The ESI-MS spectral data displayed a pseudo-molecular ion peak at m/z 271, corresponding to [M + H]^+^, in accordance with the molecular formula C_15_H_10_O_5_. The ^1^H NMR spectrum showed a singlet at δ 6.59 (1H) as well as two broad singlets at 6.46 (1H) and 6.21 (1H) attributed to the H-3, H-6, and H-8 protons, respectively, of a flavone skeleton. The ^1^H NMR spectrum also exhibited two doublets at δ 7.85 (2H, *J* = 8.4 Hz, H-2′ and H-6′) and 6.92 (2H, *J* = 8.4 Hz, H-3′ and H-5′), indicating a 4′-oxygenated phenyl group at C-2 of the flavone skeleton. In accordance with these spectral data compared to the reported values [44], compound **3** was confirmed to be 4′,5,7-trihydroxyflavone, also known as apigenin (Table 1).

Compound **4** was isolated as a bright yellow crystalline solid. The ESI-MS spectral data exhibited a pseudo-molecular ion peak [M + H]^+^ at m/z 287, in accordance with the molecular formula of C_15_H_10_O_6_. The ^1^H NMR spectrum displayed two meta-coupled (*J* = 2.0 Hz) protons at δ 6.20 (1H, d, H-6) and 6.34 (1H, d, H-8), indicating ring A of a flavone skeleton. The ^1^H NMR spectrum also exhibited two doublet signals at δ 8.00 (2H, d, *J* = 8.8 Hz, H-2′ and H-6′) and δ 6.87 (2H, d, *J* = 8.8 Hz, H-3′ and H-5′), assignable to the aromatic protons in ring B, characteristic of the 1′,4′-disubstituted flavone moiety. Based on the comparison of the above-mentioned spectral values with the published data [45], compound **4** was identified as 3,4′,5,7-tetrahydroxyflavone, also named as kaempferol (Table 1).

Compound **5** was isolated as a yellowish brown solid. The ESI-MS spectral data exhibited a pseudo-molecular ion peak [M + H]^+^ at m/z 325.029, in agreement with the molecular formula of C_20_H_20_O_4_. The ^1^H NMR spectrum demonstrated a heterocyclic ring of an isoflavane skeleton with resonances at δ 3.95 (1H, t, *J* = 10.4 Hz, H_a_-2), 4.28 (1H, ddd, *J* = 10.4, 4.0 and 2.0 Hz, H_b_-2), 3.34 (1H, m, H-3), 2.78 (1H, ddd, *J* = 15.5, 5.2 and 2.0 Hz, H_a_-4), and 2.97 (1H, dd, *J* = 15.5 and 11.2 Hz, H_b_-4). Two doublets with *J* = 8.0 Hz at δ 6.26 and 6.79 assigned to H-6 and H-5, respectively, accounted for ring A of the isoflavane skeleton. The spectrum also exhibited singlets at δ 1.36 (3H, s, H-5″) and 1.38 (3H, s, H-6″) and two doublets at δ 5.56 (1H, *J* = 10.0 Hz, H-3″) and 6.60 (1H, *J* = 10.0 Hz, H-4″), which altogether indicate a dimethyl pyran ring attached to ring A. Two doublets at δ 6.30 (1H, *J* = 2.8 Hz, H-3′) and 6.89 (1H, *J* = 8.4 Hz, H-6′) and a doublet of doublets at 6.25 (1H, *J* = 8.4 and 2.8 Hz, H-5′) indicated a 2′,4′-oxygenated aromatic ring at C-3. A comparison of these spectral data with the established values [46] confirmed the structure of compound **5** as glabridin (Table 1).

Compound **6** was obtained as an amber crystalline solid. The ESI-MS spectral data exhibited a pseudo-molecular ion peak [M + H]^+^ at m/z 195.021, in agreement with the molecular formula of C_10_H_10_O_4_. The ^1^H NMR spectrum showed a clearly identifiable signal indicative of a methoxy group at δ 3.88 (s). The spectrum also demonstrated three aromatic protons at δ 7.16 (1H, br. s, H-2), 6.78 (1H, d, *J* = 8.4 Hz, H-5), and 7.01 (1H, br. d, *J* = 8.4, H-6). Two additional proton doublets at δ 7.47 (1H, *J* = 15.2 Hz) and 6.32 (1H, *J* = 15.2 Hz), indicating the presence of trans-coupled H-1′ and H-2′ protons in the side chain of the compound, respectively. In accordance with these spectral data compared to the reported values [47], compound **6** was confirmed as 4-hydroxy-3-methoxycinnamic acid, also named as ferulic acid (Table 1).

Compound **7** was obtained as a white crystalline powder. The ESI-MS spectral data showed a pseudo-molecular ion peak [M − H]^+^ at m/z 137, in accordance with the molecular formula of C_7_H_6_O_3_. The ^1^H NMR spectrum exhibited two doublets at δ 7.85 (2H, *J* = 8.0 Hz, H-2 and H-6) and 6.78 (2H, *J* = 8.0 Hz, H-3 and H-5), characteristic of a para substituted aromatic compound. Based on the comparison of the above-mentioned spectral values with the published values [48], compound **7** was identified as 4-hydroxybenzoic acid (Table 1).

Compound **8** was obtained as a light yellow crystalline solid. The ESI-MS spectral data displayed a pseudo-molecular ion peak [M + H]^+^ at m/z 123.020, in agreement with the molecular formula of C_7_H_6_O_2_. The ^1^H NMR spectrum showed two doublets at δ 7.68 (2H, *J* = 8.8 Hz, H-2 and H-6) and 6.82 (2H, *J* = 8.8 Hz H-3 and H-5), indicative of a para substituted aromatic compound. The most de-shielded one proton singlet at δ 9.76 ppm accounted for the aldehydic proton. A comparison of these NMR data with the established data [48] confirmed the structure of compound **8** as 4-hydroxybenzaldehyde (Table 1).

### 3.2. Evaluation of Antioxidant Property

#### 3.2.1. Total Phenolic Content

The dichloromethane soluble fraction showed the highest value in the total phenolic content (289.87 ± 0.47 mg of GAE/g of dried extract) followed by the methanolic crude extract (286.91 ± 0.28 mg of GAE/g of dried extract) and ethyl acetate soluble fraction (282.61 ± 0.34 mg of GAE/g of dried extract) (Table 2). Among the fractions, the n-hexane soluble fraction exhibited the lowest phenolic content (37.07 ± 0.11 mg of GAE/g of dried extract).

#### 3.2.2. Evaluation of DPPH Scavenging Activity

The antioxidant potential of the stem extractives as well as the isolated pure compounds were compared to BHT (IC_50_ = 17.09 ± 0.01 µg/mL) (Table 2). The highest free radical scavenging activity was exhibited by the dichloromethane soluble fraction (IC_50_ = 18.83 ± 0.07 µg/mL) followed by the methanolic crude extract (IC_50_ = 20.09 ± 0.07 µg/mL). The ethyl acetate soluble fraction also showed noteworthy scavenging activity (IC_50_ = 23.36 ± 0.08 µg/mL) against the DPPH free radical. The lowest free radical scavenging activity was exerted by the n-hexane soluble fraction (IC_50_ = 53.26 ± 0.05 µg/mL).

All of the isolated pure compounds demonstrated high scavenging activity against the DPPH free radical. Compounds **1**–**7** exhibited a higher antioxidant potential than the standard BHT (IC_50_ = 6.20 ± 0.10 to 16.11 ± 0.02 µg/mL vs. 17.09 ± 0.01 µg/mL) (Table 2). Among the eight compounds, kaempferol showed the highest free radical scavenging property (IC_50_ = 6.20 ± 0.10 µg/mL) followed by glabridin (IC_50_ = 7.85 ± 0.04 µg/mL) and ferulic acid (IC_50_ = 8.31 ± 0.01 µg/mL).

### 3.3. Evaluation of Hypoglycemic Property

Both doses of the methanolic crude extract of stem reduced the blood glucose level in a statistically significant manner (Table 3). Percent reduction in the blood glucose level was found to be dose- and time-dependent. A total of 400 mg/kg of body weight dose of the methanolic crude extract showed the highest hypoglycemic activity of 52.39% after 3 h of glucose administration, whereas the standard drug glibenclamide exhibited a reduction of 59.39% in the blood glucose level (Table 3).

### 3.4. Evaluation of Antidiarrheal Activity

The reduction in the total number of diarrheal feces was found to be statistically significant at both doses of the methanolic crude extract (Table 4). Dose-dependent percent reduction in diarrhea was observed for the test groups. A total of 400 mg/kg of body weight dose of the methanolic crude extract exhibited a higher antidiarrheal activity than the other dose with a reduction of 73.68% in diarrhea, whereas the standard drug loperamide exhibited a reduction of 84.21% in diarrhea (Table 4).

### 3.5. Molecular Docking Analysis

Using many appropriate computer-based methods, molecular docking of the plant-derived bioactive constituents to the respective molecular receptors was carried out to comprehend the mechanism of the pharmacological actions of the extracts and the pure compounds obtained from *W. tinctoria*. Table 5 lists all of the docking results obtained from PyRx. Appendix A contain the amino acids responsible for the interactions with the ligands’ atoms, along with the nature of the interactions, the bond type, and the bond distances. The binding strength increases as the numerical values of the binding affinity (kcal/mol) decreases. The projected binding affinity indicated the optimum docking prediction with a null RMSD (root mean square deviation) value [44]. The ability of the isolated chemicals to block enzymes and receptors is reported as follows.

#### 3.5.1. Antioxidant Property via Inhibition of Glutathione Reductase and Urase Oxidase

All isolated compounds exerted a better binding affinity toward the glutathione reductase enzyme (−5.3 to −8.7 kcal/mol;
Table 5) and urase oxidase (−5.1 to −8.2 kcal/mol; Table 5) than the standard antioxidant drug butylated hydroxytoluene (BHT) (−5.8 kcal/mol and −5.6 kcal/mol against 3GRS and 1R4U, respectively). Glabridin (compound **5**) was the compound with the highest binding affinity to the 3GRS protein, followed by apigenin (compound **3**), kaempferol (compound **4**), liquiritigenin (compound **1**), naringenin (compound **2**), ferulic acid (compound **6**), 4-hydroxybenzoic acid (compound **7**), BHT, and 4-hydroxybenzaldehyde (compound **8**).
Figure 2
provides a summary of the glutathione reductase enzyme’s active binding sites while interacting with the isolated substances. Notably, a total of nineteen interacting sites, viz. GLY 27, SER 51, GLU 50, ASN 294, GLY 158, ARG 291, ASP 178, PRO 160, THR 57, MET 159, VAL 61, GLY 56, HIS 52, THR 156, HIS 129, ILE 26, ALA 130, VAL 49, and GLY 128 were noticed while interacting glabridin (compound **5**) with the 3GRS protein.
Appendix A
lists all of the bonding sites along with the appropriate distances of all the isolated phytocompounds while docking with the glutathione reductase enzyme.
In contrast, the compounds’ binding affinities to the urase oxidase (PDB ID: 1R4U) protein were in the following order: liquiritigenin (compound **1**) > apigenin (compound **3**) > naringenin (compound **2**) > kaempferol (compound **4**) > glabridin (compound **5**) > ferulic acid (compound **6**) > BHT > 4-hydrobenzaldehyde (compound **8**). In
Appendix A
and
Figure 3, all of the urase oxidase protein’s binding sites are listed, along with their corresponding amino acid distances from the ligands.

#### 3.5.2. Hypoglycemic Activity via Inhibition of Glucose Transporter 3

The binding affinities of the isolated phytocompounds (**1**–**8**) from the plant were −5.6 to −11.4 kcal/mol compared to the standard drug glibenclamide (−10.2 kcal/mol). The order of the docking score was as follows: compound **5** (glabridin) > glibenclamide (standard drug) > compound **1** (liquiritigenin) > compound **3** (apigenin) > compound **2** (naringenin) > compound **4** (kaempferol) > compound **6** (ferulic acid) > compound **7** (4-hydroxybenzooic acid) > compound **8** (4-hydroxybenzaldehyde) (Table 5). There was a total of twenty binding points of the GLUT 3 protein while interacting with compound **5** (glabridin). The active interacting amino acids of the protein during the interaction with compound **5** were PHE 24, TRP 28, ILE 31, VAL 67, ALA 68, PHE 70, SER 71, ARG 124, GLN 159, ILE 162, ILE 166, GLN 280, GLN 281, ILE 285, ASN 286, PHE 377, GLY 382, TRP 386, TRP 413, and ASN 413 (Appendix A). However, GLN 159, GLN 281, and GLY 382 were responsible for the conventional hydrogen bond formation. There were five hydrophobic interactions, among them, three were alkyl and two were Pi–alkyl interactions (Appendix A). Similarly, all of the bonding profiles of the other isolated compounds with the GLUT 3 protein are represented and summarized in
Appendix A
and
Appendix A.

#### 3.5.3. Antidiarrheal Activity via Inhibition of Kappa Opioid Receptor

The molecular interaction of the isolated phytocompounds (**1**–**8**) with the kappa opioid receptor (PDB ID: 6VI4) was also studied to anticipate the binding profile of these molecules. All of the isolated compound showed a lower affinity (−4.4 to 7.2 kcal/mol) toward the kappa opioid receptor than the standard drug loperamide (−7.3 kcal/mol) (Table 5). Nonetheless, compound **5** (glabridin), out of all of the isolated phytocompounds, exhibited the most affinity for the kappa opioid receptor. The active binding sites of all of the compounds are found in
Appendix A. Compound **5** (glabridin) exerted a total of nine hydrophobic interactions compared with the six hydrophobic interactions formed by loperamide (Appendix A). Similar to this, Appendix A
and
Appendix A
depict and compile the binding profiles of all of the isolated phytochemicals with the kappa opioid receptor (PDB ID: 6VI4).

## 4. Discussion

In recent years, phenolic phytochemicals have received ever-increasing attention worldwide for exerting a wide spectrum of pharmacological effects such as anticarcinogenic, antioxidant, antimicrobial, anti-inflammatory, antidiabetic, cardioprotective, antithrombotic, antidiarrheal, and antiallergenic effects [49,50,51]. Therefore, plant-derived phenolic compounds are being regarded as a thriving source of promising drug candidates with the ultimate goal of developing new medicines with more efficacy and less toxicity [52,53].

The present study demonstrated the isolation and identification of eight phenolic compounds from the dichloromethane soluble fraction of the stem extract of *W. tinctoria*, which included two flavanones, liquiritigenin (**1**) and naringenin (**2**); two flavones, apigenin (**3**) and kaempferol (**4**), an isoflavane, glabridin (**5**); a hydroxycinnamic acid, ferulic acid (**6**), and two other phenolic compounds, 4-hydroxybenzoic acid (**7**) and 4-hydroxybenzaldehyde (**8**). According to our knowledge, based on an extensive literature search, this is the first ever report of the isolation of all of these compounds from this plant species.

Excessive production of free radicals in the body triggers the development of a wide variety of diseases such as neurodegenerative diseases, cardiovascular diseases, diabetes mellitus, rheumatoid arthritis, liver diseases, obesity, and cancer [54]. Plant derived phenolic compounds can act as free radical fighters by scavenging reactive oxygen species (ROS) efficiently due to the hydroxyl groups present in their structures [55,56]. From the results of the total phenolic content and DPPH assay of different fractions of the stem extract of *W. tinctoria*, it was evident that there is an excellent correlation between the total phenolic content and free radical scavenging activity of the stem extractives. Among the fractions, the dichloromethane soluble fraction showed the highest phenolic content and lowest IC_50_ value, while the n-hexane soluble fraction showed the lowest phenolic content and highest IC_50_ value. A higher total phenolic content and lower IC_50_ value in the DPPH assay together are indicative of the abundance of phenolic phytoconstituents with significant antioxidant activity. Overall, except the n-hexane soluble fraction, all other fractions exhibited a high total phenolic content as well as notable antioxidant potential compared to the standard material BHT. Therefore, the present study suggests that *W. tinctoria* can be considered as a prospective source of natural antioxidants. As per our knowledge search, this is a baseline report of the evaluation of the antioxidant potential of the *W. tinctoria* stem.

Moreover, among the isolated compounds, except 4-hydroxybenzaldehyde (compound **8**), all of the other compounds (**1**–**7**) showed lower IC_50_ values in comparison to the standard compound BHT, which indicated the higher antioxidant potential of the compounds in comparison to BHT. The DPPH free radical scavenging activity of the isolated phenolic compounds was as follows (in descending order): kaempferol > glabridin > ferulic acid > naringenin > apigenin > liquiritigenin > 4-hydroxybenzoic acid > 4-hydroxybenzaldehyde. Similarly, the substance with the highest molecular docking score for binding to the glutathione reductase enzyme was glabridin, followed by apigenin, kaempferol, liquiritigenin, naringenin, ferulic acid, 4-hydroxybenzoic acid, BHT, and 4-hydroxybenzaldehyde.

Phenolic phytochemicals, the largest category of plant-derived secondary metabolites, have the potential to act as antioxidants in a variety of ways reported in several studies [57,58,59]. Hydroxyl groups present in the polyphenols scavenge reactive free radicals and thereby shield the biological system from oxidative stress brought on by free radicals [35,49,60]. The substances derived from the plant may demonstrate high antioxidant capabilities by directly scavenging DPPH and increasing the membrane permeability [61]. One possible explanation for these effects is that the compound interacts hydrophobically with the glutathione reductase or urase oxidase enzyme through interactions with alkyl and pi-alkyl groups (Appendix A). Kaempferol exerted substantial antioxidant properties, similar to our findings in several earlier studies. According to Tzeng et al., the 50 percent scavenging concentration of kaempferol was estimated to be 22.08 ± 3.71 μg/mL [62]. In a different investigation, Zhong et al. [63] demonstrated that the IC_50_ value of kaempferol was 16.23 μg/mL. The third hydroxyl on the C ring was suggested to be an essential factor in the antioxidant activity [63]. Kaempferol, a flavonoid containing hydroxyl at the third position on the C ring, exhibited potent radical scavenging activity. In this study, glabridin, an isoflavane, exhibited a promising antioxidant effect that was also corroborated by a number of other studies. The compound exhibits its biological properties by inhibiting the oxidation of low-density lipoprotein (LDL) and downregulating the intracellular reactive oxygen species [64,65,66]. Ferulic acid has also revealed free radical scavenging activity, which may be supported by several previous reports [67,68,69,70]. By scavenging oxidative free radicals, limiting the generation of reactive oxygen species (ROS), and participating in a number of signaling cascades, ferulic acid can carry out its antioxidant function [69,70]. Similarly, prior investigations [38,71,72,73,74,75] and the current computational profiling have also supported the possible antioxidant properties of naringenin, apigenin, liquiritigenin, and 4-hydroxybenzoic acid found in this study.

Phenolic compounds obtained from medicinal plants can also lower the blood sugar level independently or synergistically with alkaloids, steroids, terpenes, and other constituents present in the plant, and thus can be considered as an alternative method for diabetes management due to the minimal side effects and relatively low cost [76,77]. The methanolic crude extract of the *W. tinctoria* stem demonstrated statistically significant hypoglycemic effects at both doses (200 mg/kg bw and 400 mg/kg bw) at the first, second, and third hour after glucose loading. However, a 400 mg/kg bw dose of the methanolic crude extract showed the highest hypoglycemic activity after 3 h of glucose administration. The current in silico study corroborated the in vivo findings. In the molecular modeling study, glabridin had the highest hypoglycemic effect of all the extracted compounds and had an excellent binding affinity (−11.4 kcal/mol) toward the glucose transporter 3 (GLUT 3) protein. Glabridin has been shown to have hypoglycemic effects in some prior investigations, including in vivo and clinical studies [78,79]. Digestive enzymes such as α-glucosidase and α-amylase metabolize hydrolyzed carbohydrates into glucose that may be absorbed. Polyphenols, for example, kaempferol, decrease the activity of these enzymes and regulate postprandial hyperglycemia by inhibiting these metabolic enzymes [80]. Likewise, all other isolated polyphenols exerted notable binding affinity toward the GLUT 3 protein, revealing potential hypoglycemic properties. Due to the production of multiple reducing sugars via glycolysis and the polyol pathway, hyperglycemia, a factor in oxidative stress in diabetic patients, lowers the capability of the body’s natural antioxidant defense system [78]. Polyphenols significantly boost superoxide dismutase (SOD) activity and lower malondialdehyde (MDA) levels in the pancreas, kidney, and liver [78]. As all the isolated compounds from *W. tinctoria* displayed excellent antioxidative properties in both in vitro and in silico experiments, these phytocompounds might be the responsible agents for the hypoglycemic property of the plant extracts in this study. Therefore, based on the observations of the current study, *W. tinctoria* might be regarded as a potential source of hypoglycemic phytoconstituents.

Medicinal plants have been used traditionally for the management of a variety of gastrointestinal disorders, including diarrhea over the years. But most of these plants’ safety and effectiveness profiles have not been extensively examined. Therefore, this study evaluated the efficacy and safety of *W. tinctoria* as an antidiarrheal option in treating diarrheal disease. Diarrhea can be caused by an imbalance of absorption pattern in GI tract or in the smooth muscle motility of the intestinal tract [81]. Ricinoleic acid, the main component of castor oil, is said to irritate the gut wall by releasing prostaglandins, causing peristaltic motion that can lead to diarrhea [82,83,84]. The present study found statistically significant antidiarrheal activity at both doses of the methanolic crude extract of *W. tinctoria* in castor oil-induced diarrheal mice model. Phytoconstituents like glycosides, alkaloids, tannins, terpenes, flavonoids and other phenolic compounds found in medicinal plants possess strong antidiarrheal activity [85]. The present computational modeling study also found notable binding scores of the isolated flavonoids, liquiritigenin, naringenin, apigenin, and kaempferol compared to the standard drug loperamide. According to visualization and docking analysis findings, the isolated compounds interact with the targeted enzymes through a network of chemical interactions. Among the isolated compounds, glabridin exerted the highest docking score and showed a total of nine hydrophobic bonds with the kappa opioid receptor (Appendix A). Besides, liquiritigenin and naringenin also exhibited significant binding affinity toward the opioid receptor. The hydrophobic interactions of glabridin and conventional hydrogen bonds might be responsible for the strong interactions between the phytocompounds and the kappa opioid receptor. Opiates have a well-known proven history of being effective antidiarrheal treatments that decrease gastrointestinal motility by slowing down stomach motility and enhancing fluid and electrolyte absorption [86]. The flavonoid-rich plant extract from *W. tinctoria* has the potential to treat diarrhea in several mechanisms, including by preventing the production of prostaglandins. Flavonoids may inhibit the cyclooxygenase enzymes (COX-1 and COX-2) necessary for the synthesis of several inflammatory mediators, including prostaglandins [87]. By inhibiting intestinal secretion and motility, these flavonoids may be useful in treating both acute and chronic diarrhea. They may also lessen the severity of long-term inflammatory damage to the gut by shielding it from oxidative stress and maintaining mucosal function [88,89].

## 5. Conclusions

The current research has reported the isolation of eight phenolic compounds characterized as liquiritigenin (**1**), naringenin (**2**), apigenin (**3**), kaempferol (**4**), glabridin (**5**), ferulic acid (**6**), 4-hydroxybenzoic acid (**7**), and 4-hydroxybenzaldehyde (**8**) from the stem extract of *W. tinctoria*. Among all of the fractions, the dichloromethane soluble fraction exhibited the highest phenolic content (289.87 ± 0.47 mg of GAE/g of dried extract) as well as the highest DPPH free radical scavenging activity (IC_50_ = 18.83 ± 0.07 µg/mL). Furthermore, kaempferol showed the highest free radical scavenging activity (IC_50_ = 6.20 ± 0.10 µg/mL) among all of the isolated compounds, whereas the IC_50_ value of BHT was 17.09 ± 0.01 µg/mL. It has to be noted that the 400 mg/kg bw dose of the methanolic crude extract showed the highest hypoglycemic activity after 3 h of glucose administration and the highest antidiarrheal activity after 4 h of castor oil administration. Additionally, the in silico analyses corroborated the in vitro and in vivo results. Therefore, the current pharmacological analysis revealed the efficacy of *W. tinctoria* as a promising source of possible bioactive phenolic compounds that may be considered for innovative drug discovery and therapeutic advancement, focusing on the antioxidant, hypoglycemic, and antidiarrheal potentialities. Nevertheless, further research is necessary to identify more bioactive components from *W. tinctoria* that might be responsible for broader bioactivities. Furthermore, clinical studies can also be suggested based on the current evidence for ensuring the reported pharmacological effects of the plant.

## Figures and Tables

**Figure 1 molecules-27-05957-f001:**
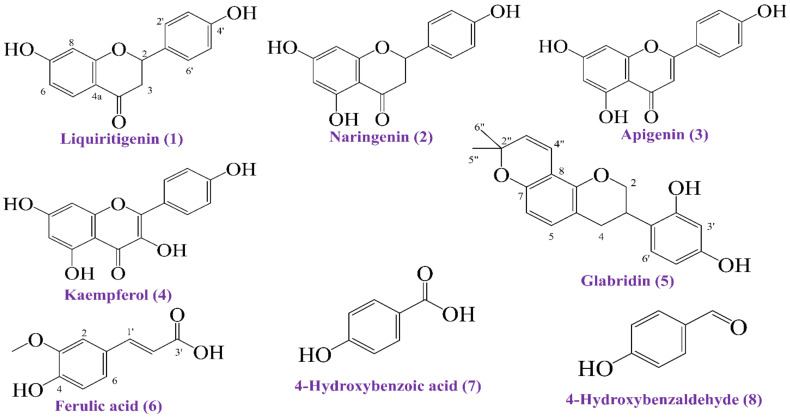
Structures of the isolated eight compounds (**1**–**8**) from the stem extract of *W. tinctoria* (compound **1**: Liquiritigenin; compound **2**: Naringenin; compound **3**: Apigenin; compound **4**: Kaempferol; compound **5**: Glabridin; compound **6**: Ferulic acid; compound **7**: 4-Hydroxybenzoic acid; compound **8**: 4-Hydroxybenzaldehyde).

**Figure 2 molecules-27-05957-f002:**
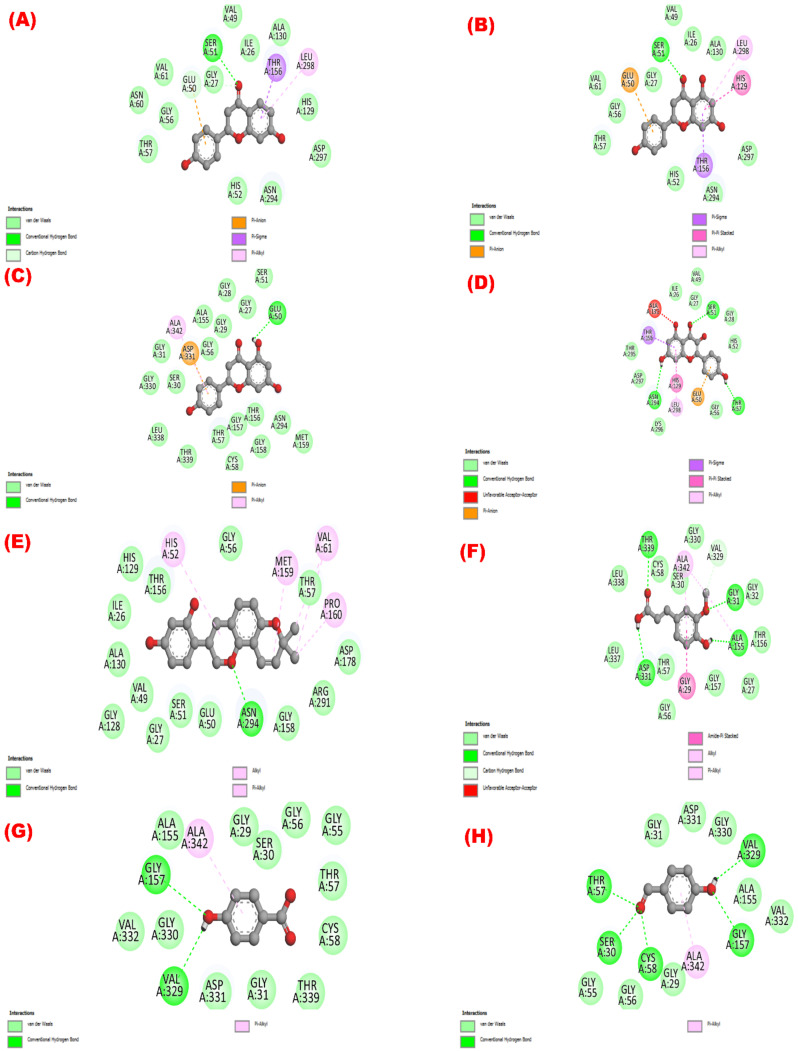
The isolated phytochemicals’ potential as antioxidants showing their 2D molecular interactions with the glutathione reductase enzyme (PDB ID: 3GRS). The 2D visual images of the molecular docking of the molecules (**1** to **8**) liquiritigenin, naringenin, apigenin, kaempferol, glabridin, ferulic acid, 4-hydroxybenzoic acid, and 4-hydroxybenzaldehyde are shown in (**A**–**H**), respectively.

**Figure 3 molecules-27-05957-f003:**
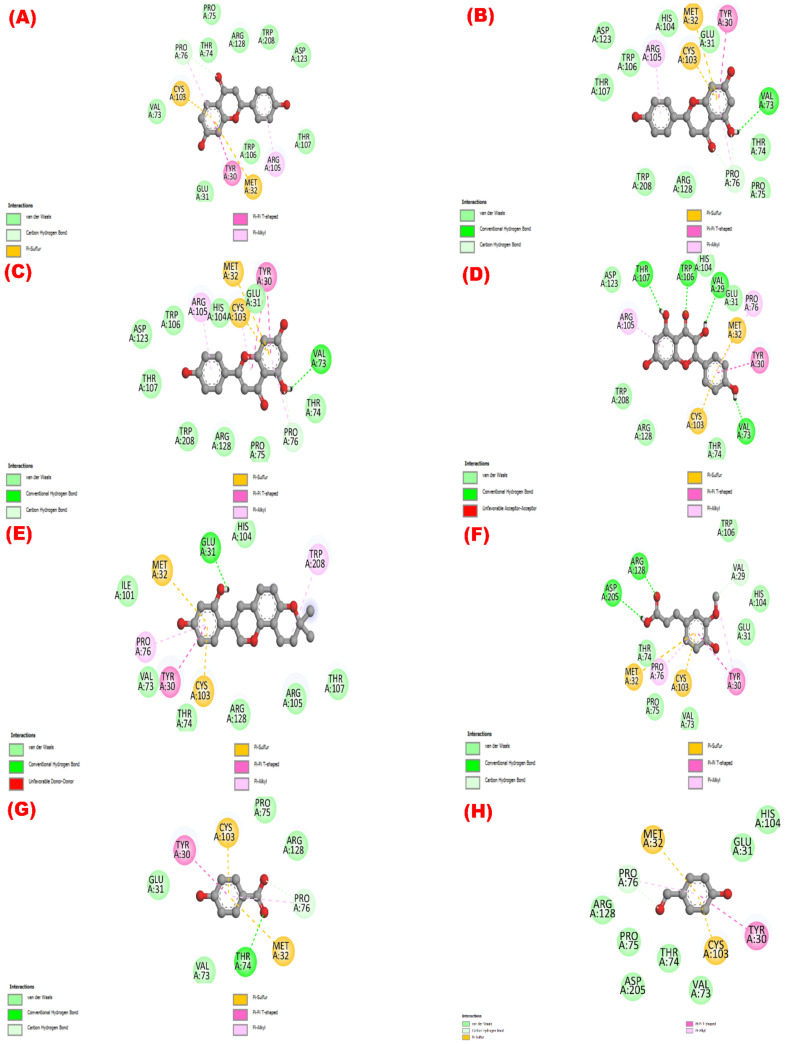
The isolated phytochemicals’ potential as antioxidants showing their 2D molecular interactions with the urase oxidase enzyme (PDB ID: 1R4U). The 2D visual images of the molecular docking of the molecules (**1** to **8**) liquiritigenin, naringenin, apigenin, kaempferol, glabridin, ferulic acid, 4-hydroxybenzoic acid, and 4-hydroxybenzaldehyde are shown in (**A**–**H**), respectively.

**Table 1 molecules-27-05957-t001:** The ^1^H NMR (400 MHz, CD_3_OD) and ESI mass spectroscopic data of the isolated compounds **1**–**8** from *W. tinctoria*.

Compounds	Physical Appearance	Mol. Formula	Mol. wt.	Pseudo-Molecular Ion Peak at m/z	^1^H NMR Data (400 MHz)	
					Position	Spectroscopic Data
Liquiritigenin (**1**)	Lemon yellow crystalline solid	C_15_H_12_O_4_	256.26	257 [M + H]^+^	H-2	δ 5.36 (1H, dd, *J* = 13.2 and 2.8 Hz)
					H_a_-3	δ 3.04 (1H, dd, *J* = 16.6 and 13.2 Hz)
					H_b_-3	δ 2.67 (1H, dd, *J* = 16.6 and 2.8 Hz)
					H-5	δ 7.72 (1H, d, *J* = 8.0 Hz)
					H-6	δ 6.48 (1H, dd, *J* = 8.0 and 2.0 Hz)
					H-8	δ 6.34 (1H, d, *J* = 2.0 Hz)
					H-6′	δ 7.33 (2H, d, *J* = 8.1 Hz, H-2′)
					H-5′	δ 6.80 (2H, d, *J* = 8.1 Hz, H-3′)
Naringenin (**2**)	Colorless crystalline solid	C_15_H_12_O_5_	272.26	273 [M + H]^+^	H-2	δ 5.35 (1H, dd, *J* = 13.2 and 2.8 Hz)
					H_a_-3	δ 2.69 (1H, dd, *J* = 17.2 and 2.8 Hz)
					H_b_-3	δ 3.10 (1H, dd, *J* = 17.2 and 13.2 Hz)
					H-8	δ 5.87 (br. s, H-6), 5.88 (br. s)
					H-2′, H-6′	δ 6.80 (2H, d, *J* = 8.0 Hz)
					H-3′, H-5′	δ 7.40 (2H, d, *J* = 8.0 Hz)
Apigenin (**3**)	Pale yellow crystalline powder	C_15_H_10_O_5_	270.24	271 [M + H]^+^	H-3	δ 6.59 (1H, s)
					H-6	δ 6.46 (1H, br. s)
					H-8	δ 6.21 (1H, br. s)
					H-2′, H-6′	δ 7.85 (2H, d, J = 8.4 Hz
					H-3′, H-5′	δ 6.92 (2H, d, J = 8.4 Hz)
Kaempferol (**4**)	Bright yellow crystalline powder	C_15_H_10_O_6_	286.24	287 [M + H]^+^	H-6	δ 6.20 (1H, d, *J* = 2.0 Hz)
					H-8	δ 6.34 (1H, d, *J* = 2.0 Hz)
					H-2′, H-6′	δ 8.00 (2H, d, *J* = 8.8 Hz)
					H-3′, H-5′	δ 6.87 (2H, d, *J* = 8.8 Hz)
Glabridin (**5**)	Yellowish brown solid	C_20_H_20_O_4_	324.38	325.029 [M + H]^+^	H_a_-2	δ 3.95 (1H, t, *J* = 10.4 Hz)
					H_b_-2	δ 4.28 (1H, ddd, *J* = 10.4, 4.0 and 2.0 Hz)
					H-3	δ 3.34 (1H, m)
					H_a_-4	δ 2.78 (1H, ddd, *J* = 15.5, 5.2 and 2.0 Hz)
					H_b_-4	δ 2.97 (1H, dd, *J* = 15.5 and 11.2 Hz)
					H-5	δ 6.79 (1H, d, *J* = 8.0 Hz)
					H-6	δ 6.26 (1H, d, *J* = 8.0 Hz),
					H-3′	δ 6.30 (1H, d, *J* = 2.8 Hz)
					H-5′	δ 6.25 (1H, dd, *J* = 8.4 and 2.8 Hz)
					H-6′	δ 6.89 (1H, d, *J* = 8.4 Hz)
					H-3″	δ 5.56 (1H, d, *J* = 10.0 Hz)
					H-4″	δ 6.60 (1H, d, *J* = 10.0 Hz)
					H-5″	δ 1.36 (3H, s)
					H-6″	δ 1.38 (3H, s)
Ferulic acid (**6**)	Amber crystalline solid	C_10_H_10_O_4_	194.19	195.021 [M + H]^+^	-OCH_3_	δ 3.88 (1H, s)
					H-2	δ 7.16 (1H, br. s)
					H-5	δ 6.78 (1H, d, *J* = 8.4 Hz)
					H-6	δ 7.01 (1H, br. d, *J* = 8.4 Hz)
					H-1′	δ 7.47 (1H, d, *J* = 15.2 Hz)
					H-2′	6.32 (1H, d, *J* = 15.2 Hz)
4-Hydroxybenzoic acid (**7**)	White crystalline powder	C_7_H_6_O_3_	138.12	137 [M − H]^+^	H-2, H-6	δ 7.85 (2H, d, *J* = 8.0 Hz)
					H-3, H-5	δ 6.78 (2H, d, *J* = 8.0 Hz)
4-Hydroxybenzaldehyde (**8**)	Light yellow crystalline solid	C_7_H_6_O_2_	122.12	123.02 [M + H]^+^	-CHO	δ 9.76 (1H, s)
					H-2, H-6	δ 7.68 (2H, d, *J* = 8.8 Hz),
					H-3, H-5	δ 6.82 (2H, d, *J* = 8.8 Hz)

**Table 2 molecules-27-05957-t002:** The total phenolic content of different fractions of the *W. tinctoria* stem, and DPPH free radical scavenging activity with IC_50_ values of various solvent fractions of the *W. tinctoria* stem and its derived phenolic compounds.

Test Material	Total Phenolic Content (mg of GAE/g of Dried Extract)	DPPH Free Radical Scavenging Activity (IC_50_, µg/mL)
MCE	286.91 ± 0.28	20.09 ± 0.22
HSF	37.07 ± 0.11	53.26 ± 0.05
DSF	289.87 ± 0.47	18.83 ± 0.22
ESF	282.61 ± 0.34	23.36 ± 0.08
ASF	278.99 ± 0.17	27.33 ± 0.08
Liquiritigenin (**1**)		15.01 ± 0.01
Naringenin (**2**)		12.11 ± 0.02
Apigenin (**3**)		13.43 ± 0.03
Kaempferol (**4**)		6.20 ± 0.10
Glabridin (**5**)		7.85 ± 0.04
Ferulic acid (**6**)		8.31 ± 0.01
4-Hydroxybenzoic acid (**7**)		16.11 ± 0.02
4-Hydroxybenzaldehyde (**8**)		18.07 ± 0.02
BHT (Standard)		17.09 ± 0.01

All values are represented as mean ± SEM (*n* = 3). MCE = Methanolic crude extract, HSF = n-Hexane soluble fraction, DSF = Dichloromethane soluble fraction, ESF = Ethyl acetate soluble fraction, ASF = Aqueous soluble fraction, and BHT = Butylated hydroxytoluene.

**Table 3 molecules-27-05957-t003:** The hypoglycemic activity of the methanolic crude extract of the *W. tinctoria* stem.

Group Code	After 1 h		After 2 h		After 3 h	
	BGL (mmol/L)	% Reduction in BGL	BGL (mmol/L)	% Reduction in BGL	BGL (mmol/L)	% Reduction in BGL
CTL	10.50 ± 0.27		9.4 ± 0.25		8.25 ± 0.17	
STD	8.45 ± 0.27 *	19.52	5.45 ± 0.49 **	42.02	3.35 ± 0.10 ***	59.39
MCE 1	9.28 ± 0.11 *	11.62	6.34 ± 0.40 **	32.55	4.60 ± 0.26 ***	44.24
MCE 2	8.73 ± 0.15 *	16.86	5.73 ± 0.16 ***	39.04	3.93 ± 0.11 ***	52.36

* *p* ˂ 0.01, ** *p* ˂ 0.001, and *** *p* ˂ 0.0001 significant compared to the control; BGL represented as mean ± SEM (*n* = 4). BGL = Blood glucose level, CTL = Negative control (1% Tween 80 in normal saline) at a 10 mL/kg bw dose, STD = Positive control (standard drug glibenclamide) at 10 mg/kg bw dose, MCE 1 = Methanolic crude extract at 200 mg/kg bw dose and MCE 2 = Methanolic crude extract at 400 mg/kg bw dose.

**Table 4 molecules-27-05957-t004:** The antidiarrheal activity of the methanolic crude extract of the *W. tinctoria* stem.

Group Code	Dose	Total Number of Diarrheal Feces (Mean ± SEM)	% Reduction of Diarrhea
CTL	10 mL/kg bw	9.50 ± 0.65	
STD	50 mg/kg bw	1.50 ± 0.29 **	84.21
MCE 1	200 mg/kg bw	3.75 ± 0.25 *	60.53
MCE 2	400 mg/kg bw	2.50 ± 0.29 **	73.68

* *p* ˂ 0.001 and ** *p* ˂ 0.0001 significant in comparison to control; *n* = 4. CTL = Negative control (1% tween 80 in normal saline), STD = Positive control (standard drug loperamide), bw = body weight, MCE 1 = Methanolic crude extract at 200 mg/kg bw dose and MCE 2 = Methanolic crude extract at 400 mg/kg bw dose.

**Table 5 molecules-27-05957-t005:** The in silico docking scores in terms of binding affinity (kcal/mol) obtained from molecular interactions of the isolated compounds from the *Wendlandia tinctoria* (Roxb.) DC. stem and the standard drugs butylated hydroxy toluene (BHT), glibenclamide, and loperamide during the interaction with glutathione reductase (PDB ID: 3GRS) and urase oxidase (PDB ID: 1R4U), glucose transporter 3 (GLUT 3) (PDB ID: 4ZWB), and kappa opioid receptor (PDB ID: 6VI4) for assessing the antioxidant, hypoglycemic, and antidiarrheal properties, respectively.

Comp. No.	Compound Name	PubChem ID	Binding Affinity (kcal/mol)			
			3GRS	1R4U	4ZWB	6VI4
**1**	Liquiritigenin	114829	**−8.2**	**−8.2**	−9.5	−6.7
**2**	Naringenin	932	**−8.2**	**−8.1**	−9.3	−6.6
**3**	Apigenin	5280443	**−8.5**	**−8.2**	−9.4	−6.7
**4**	Kaempferol	5280863	**−8.3**	**−8.1**	−9.2	−6.4
**5**	Glabridin	124052	**−8.7**	**−8.1**	**−11.4**	−7.2
**6**	Ferulic acid	445858	**−6.7**	**−6.0**	−7.1	−4.6
**7**	4-Hydroxybenzoic acid	135	**−5.9**	−5.4	−6.3	−4.5
**8**	4-Hydroxybenzaldehyde	126	−5.3	−5.1	−5.6	−4.4
Standard	BHT	31404	−5.8	−5.6		
	Glibenclamide	3488			−10.2	
	Loperamide	3955				−7.3

## Data Availability

If a reasonable request is made, the corresponding authors of this study can provide access to all of the raw data in the study.

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
