# Peer review of "Phenolic Constituents from Wendlandia tinctoria var. grandis (Roxb.) DC. Stem Deciphering Pharmacological Potentials against Oxidation, Hyperglycemia, and Diarrhea: Phyto-Pharmacological and Computational Approaches"

_molecules, 2022, doi:10.3390/molecules27185957_

Round 1

Reviewer 1 Report

-Introduction must be substantialy reduced and the informations content should be exactly at the point. also, it is important that the informations in  the paragraphs link each other: e.g. the antioxidant activity links with the polyphenols in the extracts and to the high oxidative status in diabetes. 

-it is not clear what was the state of your extract (row 140) was it solid, liquid (if so, how much dry substance does it contain?). what does it mean gm (row 145)?

-to make the reading of the text easier I propose you to eliminate the figure 1 , because the structures of the 8 identified compounds can be found on internet

-for total polyphenolic content determination method it is necessary to add the equation of the calibration curve

-to increase the impact of the research try to coroborate the in vivo results with the in silico ones. 

 - in vitro studies seem to be rather in silico studies (row 340)

-in the Results chapter all the NMR analysis results (rows 356-467) could be presented as a table, which would considerably reduce the paper extent

-in the sub-chapter Evaluation of antioxidant property, the informations in the text and the ones in the table are redundant. my advice is to keep the table

-Discussions chapter should be rewritten taking into account the recomandetion to correlate the in vivo and in silico results. 

- Conclusions must be concise and figure out some directions for the future research, e. g. medical applications

Author Response

Reviewer 1

Comments and Suggestions for Authors

Authors responses: I would like to thank you very much to reviewer 1 for your valuable time for reviewing our manuscript and providing your scholarly opinions and suggestions. All the comments and directions were warmly welcomed. We have carefully studied all these raised points and improved the manuscript accordingly. Thank you again for your kind comments and consideration.      

-Introduction must be substantialy reduced and the informations content should be exactly at the point. also, it is important that the informations in the paragraphs link each other: e.g. the antioxidant activity links with the polyphenols in the extracts and to the high oxidative status in diabetes. 

Authors' responses: Thank you very much for your comments and nice suggestions. We have significantly reduced the words from the introduction part and tried to keep the information exactly at the point. We have made concise the previous three paras (3, 4 and 5) into one para as follows:  

The progression of degenerative and chronic diseases, including cancerous growths, diabetes, immunological issues, joint inflammation, and cardiovascular and neurological disorders, has been linked significantly to oxidative stress [18]. The production of ROS by lipoxygenases, cyclooxygenases, nitric oxide synthases, and peroxidases is assumed to be triggered by hyperglycemia in diabetics [25]. Substantial antioxidant activity of phytocompounds, especially polyphenols, has been demonstrated to prevent oxidative stress by reducing ROS generation and retaining redox homeostasis [19-21]. The latest conventional anti-diabetic medications may alleviate diabetes symptoms, but they also carry a risk of severe adverse reactions, including hypoglycemic shock, edema, irregular heartbeat, and liver function, as well as stomach and respiratory problems [26]. As a result, scientists are working to identify and characterize polyphenols with potent antioxidant properties found in medicinal plants, which can be used as alternatives to traditional diabetes therapies because of their less toxicity and significant antioxidative properties [26,27]. (Lines: 73-85)

-it is not clear what was the state of your extract (row 140) was it solid, liquid (if so, how much dry substance does it contain?). what does it mean gm (row 145)?

Authors' responses: Thank you very much for your comment. The crude extract was solid state. We have revised the line. gm means gram in the line.

-to make the reading of the text easier I propose you to eliminate the figure 1, because the structures of the 8 identified compounds can be found on internet.

Authors' responses: Thank you very much for your comments. We have sketched the eight isolated compounds in Figure 1. Figure 1 will be beneficial for the readers to catch out the structural view rapidly. Besides, understanding the positions of the compounds is also important while studying the NMR spectroscopic data and physicochemical properties of the compounds. Therefore, we politely request you to consider us keeping Figure 1 in the manuscript for its potential readers’ convenience and scientific merits.   

-for total polyphenolic content determination method it is necessary to add the equation of the calibration curve

Authors' responses: Thank you very much for your comments. We have added the information as follows:

The TPC test was carried out 3 times. Equations of the calibration curve of gallic acid were as follows (Lines: 218-222)-

1st time experiment: y = 0.0123x - 0.19; R2 = 0.9979

2nd time experiment: y = 0.0137x - 0.2235; R2 = 0.9977

3rd time experiment: y = 0.014x - 0.241; R2 = 0.9978

-to increase the impact of the research try to coroborate the in vivo results with the in silico ones. 

Authors' responses: Thank you very much for your comments. We have revised the discussion part and adjusted the in vivo results with in silico results.

 - in vitro studies seem to be rather in silico studies (row 340)

Authors' responses: Thank you very much for your comments. We have revised the lines and corrected.

-in the Results chapter all the NMR analysis results (rows 356-467) could be presented as a table, which would considerably reduce the paper extent

Authors' responses: Thank you very much for your excellent suggestion. We have tabulated the mass and NMR data in Table 1 and considerably reduced the paper's extent.

-in the sub-chapter Evaluation of antioxidant property, the informations in the text and the ones in the table are redundant. my advice is to keep the table

Authors' responses: Thank you very much for your comments. We have kept the data in Table 2 and explained the data in the text as briefly as possible.

-Discussions chapter should be rewritten taking into account the recomandetion to correlate the in vivo and in silico results. 

Authors' responses: Thank you very much for your comments and nice suggestions. We have the discussion part carefully and correlated the in vivo findings with the in-silico results. We have assumed several mechanisms of action of these in vivo biological activities with the help of in silico findings. We added several more lines according to your suggestions to improve the discussion part and correlated the in vivo results with in silico prediction as follows:

According to visualization and docking analysis findings, the isolated compounds interact with the targeted enzymes through a network of chemical interactions. Among the isolated compounds, glabridin exerted the highest docking score and showed a total of nine hydrophobic bonds with the kappa opioid receptor (Table S7). Besides, liquiritigenin and naringenin also exhibited significant binding affinity towards the opioid receptor. The hydrophobic interactions of glabridin and conventional hydrogen bonds might be responsible for the strong interactions between the phytocompounds and the kappa opioid receptor. Opiates have a well-known proven history of being effective anti-diarrheal treatments that decrease gastrointestinal motility by slowing down stomach motility and enhancing fluid and electrolyte absorption. (Lines: 679-689)

Conclusions must be concise and figure out some directions for the future research, e. g. medical applications

Authors' responses: Thank you very much for your excellent suggestions. We have made the conclusion concise and directed some future research, like clinical studies, based on the current evidence for ensuring these reported pharmacological effects of the plant.

Reviewer 2 Report

The Article entitled ‘Phenolic Constituents From Wendlandia tinctoria (Roxb.) DC. Stem Deciphering Pharmacological Potentials Against Oxidation, Hyperglycemia and Diarrhea: Phyto-Pharmacological and Computational Approachesdescribes antioxidant, hypoglycemic, and antidiarrheal properties, and as well isolations of secondary metabolite from Wendlandia tinctoria (Roxb.) DC. (Family: Rubiaceae).

The article is well described with sufficient details. The work performed is of good quality and well written.

I recommend this article to be accepted (after minor revision) for publication in your respected journal Marine Drugs.

Author Response

Authors' responses: I would like to thank you very much to reviewer 2 for your valuable time in reviewing our manuscript and providing your scholarly opinions and feedback. We are very grateful to you for your kind comments and motivational appreciation. Thank you very much again for your kind approval of the manuscript.   

Reviewer 3 Report

The manuscript Phenolic Constituents From Wendlandia tinctoria (Roxb.) DC. Stem Deciphering Pharmacological Potentials Against Oxidation, Hyperglycemia and Diarrhea: Phyto-Pharmacological and Computational Approaches is interesting and has scientific merits

Among these points:

Major points:
1- The Scheme of isolation for the main components are highly misleading and should be given in the supplementary data as a schematic sketch with all weights
2- The authors should notice that the plant was identified with many subspecies and varieties, therefore they should confirm to which level they identified
3- The docking part should be revised, one or two figures are more than enough to represent the highest binging compound and the rest should be shifted into supplementary they should validate their work by docking known inhibitor to check the results

Minor points:
1- It will better if tables 1-3 could be modified into more visual friendly figures to help the readers getting the main results instead of these large number of digits.
2- In the intro part facts about the diseases are not required, only related to the plant itself will be more than enough

3- The spectral data of the known compounds should be shifted into the supplementary data

4- The references should be reduced, it is not accepted to cite more than 90 references for a research article  

Author Response

Reviewer 3

The manuscript Phenolic Constituents From Wendlandia tinctoria (Roxb.) DC. Stem Deciphering Pharmacological Potentials Against Oxidation, Hyperglycemia and Diarrhea: Phyto-Pharmacological and Computational Approaches is interesting and has scientific merits

Authors' responses: I would like to thank you very much to reviewer 3 for your valuable time for reviewing our manuscript and providing your scholarly opinions and suggestions. All the comments and directions were warmly welcomed. We have carefully studied all these raised points and improved the manuscript accordingly. Thank you again for your kind comments and consideration.      

Among these points:

Major points:
1- The Scheme of isolation for the main components are highly misleading and should be given in the supplementary data as a schematic sketch with all weights

Authors' responses: Thank you very much for your comments and nice suggestions. We have revised the isolation part and made a schematic diagram which has been added in the supplementary file as follows:

Figure S1: The schematic diagram of whole phytochemical isolations.

2- The authors should notice that the plant was identified with many subspecies and varieties, therefore they should confirm to which level they identified

Authors' responses: Thank you very much for your comments. We have identified the plant species as Wendlandia tinctoria var. grandis which has been updated in the title and in the manuscript's revised version. The species has been identified and validated by a specialist from the Bangladesh National Herbarium (BNH), Mirpur, Dhaka, validated the plant's taxonomical identification. The herbarium received a voucher specimen, and DACB 56484 was assigned its accession number.

3- The docking part should be revised, one or two figures are more than enough to represent the highest binging compound and the rest should be shifted into supplementary they should validate their work by docking known inhibitor to check the results.

Authors' responses: Thank you very much for your comments and suggestions. We have shifted the figures obtained from the molecular docking of these isolated compounds from manuscript main text to supplementary file. The inhibitors or approved available drugs were also docked and compared the docking results of the isolated compounds with these known inhibitors.

Minor points:
1- It will better if tables 1-3 could be modified into more visual friendly figures to help the readers getting the main results instead of these large number of digits.

Authors' responses: Thank you very much for your suggestions. We have sketched two more figures for easier representation of these results alongside some tabular representation for readers’ convenience.

Figure 2. Percentage (%) of reduction blood glucose level at different time interval (1h, 2h, and 3 h). [Here, STD = glibenclamide, MCE 1 = Methanolic Crude Extract of W. tinctoria at 200 mg/kg bw and MCE 2 = Methanolic Crude Extract of W. tinctoria at 400 mg/kg bw]. 

Figure 3. Percentage (%) of reduction of diarrheal level in mice model. [Here, STD = loperamide, MCE 1 = Methanolic Crude Extract of W. tinctoria at 200 mg/kg bw and MCE 2 = Methanolic Crude Extract of W. tinctoria at 400 mg/kg bw].

2- In the intro part facts about the diseases are not required, only related to the plant itself will be more than enough

Authors' responses: Thank you very much for your comments and suggestions. We have eliminated these disease parts and kept one para briefly as per the suggestion of reviewer 1 to integrate and rationalize these three biological activities of the study.

3- The spectral data of the known compounds should be shifted into the supplementary data

Authors' responses: Thank you very much for your comments. As per the suggestion of reviewer 1, we have tabulated the isolated phytocompounds in a table and reduced the paper extent.

4- The references should be reduced, it is not accepted to cite more than 90 references for a research article  

Authors' responses: Thank you very much for your comments. We have reduced the number of citations. The current number of citations is below 90.

Round 2

Reviewer 1 Report

In the chapter Results Figure 2 point out well the hypoglicemic effect of your extracts. It is not necessarly to keep the Table 3 also with the same data. 

Also, Figure 3 is enough to show the results, no need of Table 4.

Keep a single way to present your results either table or figure!

Author Response

Reviewer 1

In the chapter Results Figure 2 point out well the hypoglicemic effect of your extracts. It is not necessarly to keep the Table 3 also with the same data. 

Also, Figure 3 is enough to show the results, no need of Table 4.

Keep a single way to present your results either table or figure!

Authors’ responses: We would like to thank you very much for your valuable time in reviewing the manuscript for the second time. Thank you very much for your insightful comments and kind suggestions. We have kept the tables and removed figures 2 and 3. If we exclude the tables, some essential values might be missing. That’s why we have kept tables 3 and 4 and included two more columns to add the % reduction of blood glucose level and % reduction of diarrhea, respectively, in Tables 2 and 3. 

Reviewer 3 Report

The authors responded positively with most of the raised points

It could be accepted in the present form 

Author Response

Reviewer 3

The authors responded positively with most of the raised points

It could be accepted in the present form 

Authors' responses: I would like to thank you very much for your valuable time in reviewing our manuscript for the second time. Thank you very much again for your kind approval of our manuscript for acceptance. 
